# Effect of LNR-g-MMA on the Mechanical Properties and Lifetime Estimation of PLA/PP Blends

**DOI:** 10.3390/polym15071712

**Published:** 2023-03-29

**Authors:** Kraiwut Wisetkhamsai, Weerawat Patthaveekongka, Wanvimon Arayapranee

**Affiliations:** 1Department of Chemical Engineering, Faculty of Engineering and Industrial Technology, Silpakorn University, Muang, Nakorn Pathom 73000, Thailand; wisetkhamsai_k@su.ac.th (K.W.); patthaveekongka_w@silpakorn.edu (W.P.); 2Department of Chemical Engineering, College of Engineering, Rangsit University, Muang, Pathumthani 12000, Thailand

**Keywords:** blends, polylactide, polypropylene, compatibilizer, activation energy

## Abstract

Polylactide (PLA) polymer, polypropylene (PP) polymer, and a PLA/PP (70:30 wt%) blend, with liquid natural rubber−graft−methy methacrylate (LNR−g−MMA) of 0.0, 2.5, 5.0, and 10.0 phr as compatibilizers, were prepared by internal mixing and compression molding. The effect of LNR-g-MMA content on the morphology, mechanical properties, water absorption, thermal degradation, and a lifetime of blends based on PLA and PP was investigated. Scanning electron microscopy (SEM) revealed that the PLA/PP blend underwent phase separation, and the presence of LNR−g−MMA in the PLA/PP blend showed a more homogenized and refined blend morphology. Hence, the addition of LNR−g−MMA was used as a compatibilizer to induce miscibility in the PLA/PP blend. The values of tensile strength, elongation at break, and impact strength of the polymer blends increased, whereas water absorption values decreased with increased LNR−g−MMA content. Thermal degradation kinetics was studied over a temperature range of 50–800 °C with multiple heating rates. The results demonstrated that the thermal stability of blends without LNR-g-MMA was greater than that of blends with LNR−g−MMA and that the thermal stability decreased with increasing LNR−g−MMA content. The activation energy (E_a_) was calculated by using the Kissinger–Akahira–Sunose method. The E_a_ value of PLA was much lower than that of PP, and incorporating PP in the PLA matrix increased the E_a_. The addition of LNR−g−MMA to the PLA/PP blend decreased the E_a_. The lifetime of PLA/PP blends was reduced with the addition of LNR−g−MMA.

## 1. Introduction

In recent years, plastic waste has been increasing. The majority of plastic waste is generated from the petroleum industry, which cannot be replaced and degraded naturally. The disposal methods of these plastics are land-filled or incinerated, and some of them can be renewable [1,2,3]. They are one of the causes of severe environmental pollution. Therefore, this research was focused on the use of biodegradable polymers. Polylactide (PLA), a biodegradable polyester obtained by the synthesis of lactic acid produced via the fermentation of carbohydrate crops, has a good tensile strength and stiffness compared to general commercial polymers. Therefore, PLA is widely used as a packaging material, such as high−value films, coated paper packaging, and food and beverage containers [4,5]. However, the use of PLA has several limitations: they are brittle, possess a low impact resistance, and are costly [4,6]. Hence, improving the properties or limitations of general polymers can be undertaken through many methods, such as plasticization, chemical modification, and blending with other polymers, which is the most popular method. The blending of PLA with other tough polymers and thermoplastic elastomeric polymers, such as linear low density polyethylene (LLDPE) [6], poly(propylene carbonate) (PPC) [7], epoxidized natural rubber (ENR) [8], poly(bulylene succinate−butylene terephthalate) (PBST) [9], polyhydroxybutyrate (PHB) [10], or polypropylene (PP) [2,3,11,12,13,14,15], is regarded as a useful and economical way to overcome these limitations. The advantages of polymer blends result in new materials with a wide range of properties at a lower cost and higher efficiency compared to other methods. PP has outstanding mechanical properties, such as toughness, non−brittleness, and impact resistance, it is non-toxic and is cheap [16,17]. Thus, the PP/PLA blend is quite interesting and can create materials with an improved moisture resistance, mechanical properties, impact resistance, and flexural modulus. However, the polymer blend of PLA/PP is immiscible due to PLA and PP being polar and non−polar substances, respectively [18]. This results in poorer mechanical properties. The compatibility of PLA/PP depends on interfacial tension and adhesion between the two phases. A compatibilizer can be added to improve the compatibility of immiscible blends [2,3,11,12,14,18]. Numerous published research studies [3,12,14,18] have been reported on PLA/PP blends using polypropylene−g−maleic anhydride (PP−g−MAH) as a compatibilizer to enhance the miscibility and decrease the interfacial tension between PLA and PP. Choudhary et al. [3] prepared a blend of PLA and PP at various ratios (PLA:PP = 90:10, 80:20, 70:30, and 50:50) using reactive compatibilizers PP−g−MAH and glycidyl methacrylate (GMA) to induce miscibility. They found that adding PP−g−MAH, a compatibilizer, decreased the dispersed phase’s domain size and increased the mechanical characteristics of PLA/PP/ PP−g−MAH. Pivsa−Art et al. [12] reported polymer blends with PLA/PP ratios of 80:20 and 20:80, and 1, 3, and 5 wt% PP−g−MAH. Thermal, mechanical, and morphological analyses were performed on dry blend injection−molded samples. The thermal investigation showed that PP−g−MAH did not affect the polymer components’ crystalline melting temperature, but it improved impact strength. Yoo et al. [18] studied the effects of compatibilizers and hydrolysis on the tensile and impact strength, interfacial tension, and morphology of PP/PLA (80:20) blends with PP−g−MAH and styrene−ethylene−butylene−styrene−g−maleic anhydride (SEBS−g−MAH) as compatibilizers. They found that SEBS−g−MAH did not affect the compatibility of PP/PLA (80:20) blends. The impact strength of PP/PLA (80:20) blends increased with SEBS−g−MAH, suggesting that PP−g−MAH may operate as an impact modifier. Bijarimi et al. [2] presented a melt blend of PLA/PP and PLA/PP with liquid natural rubber (LNR) compatibilizer in a ratio of 90:10 for PLA/PP and 90:10:10 for PLA/PP/LNR. They discovered that the elongation at break, flexural, and notched impact strength rose dramatically for the LNR−compatible PLA/PP blend. Ebadi−Dehaghani et al. [11] produced a compatibilized terpolymer of ethylene, butyl acrylate, and glycidyl methacrylate, and non−compatibilized blends of PP and PLA with different nanoclay particle-containing compositions. According to their findings, adding more PLA to the blends increased their tensile modulus and strength while decreasing their elongation at break. The decrease in the size of the disperse phase and increase in the polydispersity of the droplet size in compatibilized PP/PLA blends, as compared to those for uncompatibilized blends, correspond to the effect of the terpolymer compatibilizer on the interfacial interaction between the phases, resulting in a smaller droplet size and a higher polydispersity. The relationship between immiscible PLA/PP blends of different compositions and the compatibilizer loading via melt mixing has been reported in a few studies. The effect of compatibilizer loading on the blends’ properties was also investigated. However, the addition of more than 20–30% compatibilizer into PLA matrix can cause the phases between the polymer blend to separate [19]. Kang et al. [20] prepared PLA/PP blends of different compositions with 0, 2.5, 5.0, 7.5, and 10.0 wt% EBA−GMA (ethylene-butyl acrylate−glycidyl methacrylate terpolymer) used as a compatibilizer. They reported that the tensile strength of the PLA/PP (70:30 wt%) blends reached a maximum for 2.5 wt% EBA−GMA, and impact strength increased with increasing EBA−GMA content. Jariyakulsith and Puajindanetr [21] studied the relationship between compatibilizer and yield strength of PLA/PP blends at 70:30, 50:50, and 30:70 ratios. In addition, PP−g−MAH was added as a compatibilizer at 0.3 and 0.7 phr (parts per hundred) of PLA/PP resin and dicumyl peroxide (DCP) as an initiator at 0.03 and 0.07 phr. They discovered that a 70:30 blend of 0.3 phr PP−g−MAH and 0.03 phr DCP provided the highest yield strength of 27.68 mPa.

The preparation of liquid natural rubber (LNR) from the degradation of natural rubber (NR) latex is related to the breaking of the natural rubber (NR) chain via direct scission of the C−C or C=C bonds of the polyisoprene backbone. Moreover, LNR with a similar microstructure of NR having a molecular weight below 20,000 g/mol becomes an attractive material for enhancement of further modifications. The graft copolymers consist of one or more side chains structurally distinct from the main polymer chains or backbone. Graft copolymerization is an established technique for modifying core–shell particles, which typically consist of a hard plastic shell phase made of polystyrene or poly(methyl methacrylate) and an elastic rubber core phase made of polybutadiene, poly(butyl acrylate), or polyisoprene. The core phase is capable of cavitating and promoting matrix shear yielding. The shell phase interacts with the matrix via physical or chemical interaction, enabling good particle dispersion in the blends [22,23]. Graft copolymer synthesis has been performed in solution [24], solid [25], and latex phases [26]; however, latex modification may be the most cost-effective and practical method. The graft copolymer of methyl methacrylate onto LNR (LNR−g−MMA), comprising an inner soft polymer sphere (LNR), i.e., the “core,” and an outer rigid polymer (PMMA), i.e., the “shell,” can be expected to have better impact resistance properties. Lee and Chang [27] reported that the core–shell elastomeric particles maintained their structural integrity while remaining well distributed throughout the PVC matrix after the melt blending process. In this article, we synthesized graft copolymer particles with a core–shell structure made of MMA and a rubbery phase of LNR. These core–shell particles are environmentally friendly composite materials and employed as a compatibilizer for PLA/PP blends. Nevertheless, no studies have reported the effect of grafting MMA onto LNR particles (LNR−g−MMA) as a compatibilizer or impact modifier in PP/PLA blends. The thermal stability and lifetime of degradable polymer blends must be realized to develop innovative methods of waste disposal, service life, and recycling for these materials on an industrial scale to alleviate the worldwide energy crisis and environmental pollution problems. Many studies [16,28,29,30] have been conducted and reported on the kinetics of thermal degradation of PLA and/or its blends. To date, no studies have investigated how adding LNR−g−MMA as a compatibilizer affects the thermal degradation behavior and the lifetime of PP/PLA blends using TGA. Thermogravimetric analysis (TGA) is typically performed to determine the thermal degradation kinetics. The activation energy (E_a_) can be calculated from degradation kinetics to estimate the polymer blend lifetime at different temperatures. The varied activation energy explains the thermal degradation process’s complexity at different conversion degrees. The activation energies are derived from several isoconversional kinetic approaches that collect thermal degradation data at various temperature data and determine the activation energy distribution as the reaction. In general, the activation energy of any solid phase reaction is a function of the temperature−dependent properties of the reaction media. This study investigates the Kissinger–Akahira–Sunose (KAS) isoconversional model.

Overall, due to the brittleness of PLA, it can be blended with PP at ratios of 70:30 wt% [20,21], yielding higher mechanical properties. It is well known that even a small quantity of compatibilizer can cause significant changes to a material’s mechanical properties [19]. LNR−based latex particles with a sufficiently thick shell, comprising an inner soft polymer sphere, the “core”, and an outer hard polymer (PMMA), the “shell,” give a free−flowing powder, which is usually employed in continuous extrusion processes for the preparation of polymer blends. 

This work aims to improve the compatibility between two types of polymer blends, PLA/PP (70:30 wt%), by adding a third component of LNR−g−MMA to act as the compatibilizer agent. The first portion of this study focused on synthesizing LNR−g−MMA using a redox initiator for core–shell−shaped particles. Later, the PLA/PP (70:30 wt%) blends with a certain amount of LNR−g−MMA (0, 2.5, 5.0, and 10.0 phr) were prepared by melt blending in an internal mixer, and molded by compression molding. The effect of LNR−g−MMA content on the morphological, mechanical, water absorption, thermal degradation behavior, and thermal stability of PLA/PP blends was investigated. The model-free kinetic method was used to estimate the degradation activation energy (E_a_), which was used to predict the lifetime of PLA, PP, and PP/PLA blends without and with LNR−g−MMA content as a compatibilizer.

## 2. Experimental

### 2.1. Materials

For this study, PLA (Ingeo^TM^ Biopolymer 2003D) was purchased from Nature works LLC, Plymouth, MN, USA. It has a melt flow index of 5.50 g/10 min (180 °C/2.16 kg) and a density of 1.24 g/cm^3^. The melting temperature of PLA was around 210 °C. Regarding the PP, PP (EL−PRO^™^ P600F) with a density of 0.9 g/cm^3^ and a melt flow index of 2.1 g/10 min (230 °C/2.1 kg) purchased from Nature works LLC, Plymouth, MN, USA, from SCG Chemicals (Bangkok, Thailand). LNR was synthesized with oxidation degradation, then the obtained LNR was grafted with MMA using an emulsion polymerization technique. The obtained LNR−g−MMA is in the form of a powder and was used as a compatibilizer between PLA and PP.

### 2.2. Preparation of LNR-g-MMA

A total of 227 g of the NR latex, having a 60% DRC (dry rubber content), was mixed with Emulvin WA (C_24_H_22_O_2_), used as a surfactant, and distilled water in a flask equipped with a condenser and stirrer. The NR latex was acidified by formic acid to pH = 5, followed by successive treatment with 0.3 mol of H_2_O_2_ and 0.02 mol of NaNO_2_ at 65 °C for 12 h. Following the reaction, 5% sodium sulfite solution was added to the mixture to remove excess H_2_O_2_. Then, the latex was left for 24 h at room temperature to obtain LNR with M_n_ = 17,139 g/mol.

Figure 1 shows oxidative degradation in producing hydroxyl−terminated LNR. Free radicals formed from the bond cleavage will lead to the degradation of NR [31].

The graft copolymerization was then continued by adding 2.26 mol of MMA. The LNR seed latex was swollen with the monomer mixture for 24 h at room temperature before adding the redox initiation system, consisting of 0.02 mol of cumene hydroperoxide and 0.02 mol of tetraethylene pentamine. The polymerization reaction was performed at a stirring speed of 400 rpm for the desired time of 8 h at 70 °C. The polymerization was stopped by adding phenol. The post−treatment included the coagulation of polymer latex and washing with deionized water. The gross polymer was recovered and dried to a constant mass in a vacuum oven at 40 °C.

The proposed mechanism of preparing LNR−g−MMA initiated with cumene hydroperoxide and tetraethylene pentamine is shown in Figure 2. The cumyloxyl [32], dimethylphenyl carbinol, or hydroxyl [33] radicals (generated from cumene hydroperoxide) tend to favor the abstraction of α−methylenic hydrogen from the allylic carbon in the polyisoprene backbone, leading to the formation of polyisoprene radicals that initiate MMA to form the graft copolymers.

Ungrafted LNR was washed out for 24 h in a soxhlet extractor using 60–80 °C boiling point petroleum ether. The residue was dried to a constant weight in an oven at 40 °C under vacuum for 24 h. The residue was extracted in acetone to remove the free polymer. The weight of the initial sample and extracted samples were measured for the determination of the graft copolymer and free polymer contents. In this system, the grafting efficiency (GE) was calculated using the following relationship:(1)$$\mathrm{Grafting}\;\mathrm{efficiency}\;(\mathrm{GE},\;\%)=\frac{\mathrm{total}\;\mathrm{weight}\;\mathrm{of}\;\mathrm{MMA}\;\mathrm{grafted}}{\mathrm{total}\;\mathrm{weight}\;\mathrm{of}\;\mathrm{MMA}\;\mathrm{polymerize}\;d}\times 100$$

After solvent extraction, FTIR analysis was performed on the graft copolymer. Figure 1 depicts the FTIR spectrum of the residue (the graft copolymer). At wavenumbers 837, 1729, and 3400 cm^−1^, three peaks are attributable to the R_2_C=CHR bending of isoprene and C=O stretching of ester groups of MMA, and the stretching mode of the hydroxyl groups, respectively. This demonstrates that the graft copolymer was formed during the grafting of MMA onto LNR.

Using SEM, the morphology of the particles was investigated. The grafting of MMA onto LNR is an emulsion copolymerization of core–shell type. The gloomy domain is LNR, and the bright domain is PMMA. The surface of the LNR is smooth, as shown in Figure 2a. A rough layer on the surface grafted onto the surface of the LNR particles is present. The LNR−g−MMA particles (86% GE) were composed of the LNR core (gloomy domain) and the compatibilized PMMA shell (bright domain), which was able to completely cover the gloomy spheres (Figure 2b). Figure 1 and Figure 2 provide evidence of MMA grafting on the LNR backbone chain in the core–shell latex particles.

### 2.3. Preparation of Injection Molding

Polymer blends of PLA, PP, and PLA/PP in a ratio of 70:30 wt%, with LNR−g−MMA of 0.0, 2.5, 5.0, and 10.0 phr as a compatibilizer, were prepared using an internal mixer (MX105-D40L50, Charoentut Model, Charoentut, Co. Ltd., Samutprakarn, Thailand) at 180 °C. Blending was carried out at a rotor speed of 70 rpm for 13 min. Prior to mixing, PLA, PP, and LNR−g−MMA as a compatibilizer were dried in an oven at 80 °C under vacuum for 4 h. After blending, a compression molding machine was used to mold the samples into dumbbell and bar shapes.

### 2.4. Characterization

#### 2.4.1. FTIR Spectroscopy

The chemical structure of rubber samples was determined by attenuated total reflectance-Fourier transform infrared (ATR−FTIR) spectroscopy (Perkin Elmer Frontier, Shelton, CT, USA). The samples were analyzed in transmittance mode within the range 600–4000 cm^−1^.

#### 2.4.2. Morphology

The morphology of rubber particles was analyzed using a field−emission scanning electron microscope (FE−SEM, Tescan, Mira3, Brno, Czech Republic). Samples were coated with a thin gold layer and detected by backscattered electrons with an acceleration of 5 kV.

#### 2.4.3. Mechanical Properties

Tensile testing was determined according to ASTM D638 using a universal tensile testing machine (EZ−LX, Shimadzu, Tokyo, Japan). Tensile properties were obtained at room temperature with a crosshead speed of 50 mm/min and a load cell of 10 N. Dumbbell shaped specimens of all the blending samples were prepared following type IV, a thickness of 3 mm, an overall width of 25.7 mm, an overall length of 115 mm, and a gage width of 6 mm. The tests were conducted to determine the properties of the materials, such as tensile strength and elongation at break. The reported values were the average of six samples.

Notched Izod impact testing was determined following ASTM D256. Samples for the Izod impact test were prepared using the compression molding process. The dimensions of a standard specimen for ASTM D256 are 63.5 × 12.7 × 3.2 mm (thickness × width × length). All specimens were notched prior to testing. The impact tester (GOTECH, GT−7045−MD, Taichung, Taiwan) was used with a pendulum energy of 2.7 J at room temperature. The impact strength was reported as energy lost per unit cross-section area at the note (J/m^2^). The test result was an average of four specimens for each condition.

#### 2.4.4. Water Absorption

Water absorption was determined according to ASTM D570. Following the moisture absorption test, the dried samples were submerged in distilled water for a period of 24 h. Every 24 h, each specimen was taken out of the distilled water and dry-wiped with a clean and dry cloth. A precise 4−digit balance was used to determine the content of water absorbed. The percentage water absorption was determined by the difference between the weights of fully saturated specimens to the weights of dry specimens, using the following Equation (2):(2)$$\%W_{w}=\frac{w_{2}-w_{1}}{w_{1}}\times 100$$
where %W_w_ is the percentage of water absorption, w_2_ represents the weight of the fully saturated sample (g), while w_1_ denotes the weight of the dry specimen (g). The test result was the average weight of five specimens for each condition.

#### 2.4.5. Thermogravimetric Analysis (TGA)

Thermogravimetric analysis (TGA) was performed using a Pyris 1 TGA (Perkin Elmer, Pyris 1, Danbury, CT, USA) thermogravimetric analyzer. Samples of 5.0–10.0 mg were weighed in alumina crucibles. TGA was conducted under air with a flow rate of 20 mL/min. Blend samples were subjected to heating rates of 10, 20, 30 and 40 °C/min, between 50 and 800 °C, to evaluate the thermal stability and degradation kinetics.

### 2.5. Kinetic Analysis

Determination of the kinetic parameters of thermal degradation can be calculated with data obtained from the TGA analysis. In general, these parameters were evaluated by a model−free kinetic method. The rate of reaction (dw/dt) for thermal degradation can be calculated as follows:(3)$$\frac{\mathrm{dw}}{\mathrm{dt}}\;=\;k(T)\cdot f(w)$$
where k(T) is the reaction rate constant, w is the conversion degree during the degradation reaction, f(w) is the reaction model, and T is the absolute temperature (K). The Arrhenius equation can describe the temperature dependence of the rate constant.
(4)$$k(T)\;=\;A\cdot \exp(-\frac{E_{a}}{\mathrm{RT}})$$
where A is the pre−exponential factor, E_a_ is the activation energy, and R is the gas constant (8.314 J/mol · K). For non-isothermal TGA, the conversion degree (w) can be determined as the ratio of the actual weight loss at time (t) to the total weight loss corresponding to the decomposition process.
(5)$$w(-)\;=\;\frac{w_{i}{\;-\;w}_{t}}{w_{i}{\;-\;w}_{f}}$$
where w_i_, w_t,_ and w_f_ are the actual weight of the sample at the initial weight, weight at time (t), and final weight of a sample, respectively. By combining Equations (3) and (4), the following Equation (6) can be obtained:(6)$$\frac{\mathrm{dw}}{\mathrm{dt}}\;=\;A\cdot \exp(-\frac{E_{a}}{\mathrm{RT}})\cdot f(w)$$

For non-isothermal conditions, in which the samples are at a heating rate (β = dT/dt):(7)$$\frac{\mathrm{dw}}{\mathrm{dt}}\;=\;\frac{\mathrm{dw}}{\mathrm{dT}}\times \frac{\mathrm{dT}}{\mathrm{dt}}\;=\;\beta \frac{\mathrm{dw}}{\mathrm{dT}}$$

The rate of degradation (dw/dt) equal to β(dw/dT) is arranged and modified as follows:(8)$$\frac{\mathrm{dw}}{\mathrm{dT}}\;=\;\frac{A}{\beta }\cdot \exp(-\frac{E_{a}}{\mathrm{RT}})\cdot f(w)$$

Following this rearrangement, Equation (9) is the fundamental form of kinetic analytical methods, which is used to evaluate the kinetic parameters for the thermal degradation data.

One of the most popular methods is the Kissinger–Akahira–Sunose (KAS) method, which is the isoconversional model−free kinetic method with multiple heating rates. It is generally known that the KAS method can be used to evaluate the activation energy (E_a_).
(9)$$\ln\frac{\beta }{T_{}{}^{2}}\;=\;\mathrm{constant}\;-\;\frac{E_{a}}{\mathrm{RT}}$$

The same degree of conversion exists at different heating rates. Thus, the activation energy can be calculated from the slope of a linear plot of ln (β/T^2^) against 1000/T at the constant conversion degree (w).

The kinetic analysis of the non-isothermal data was obtained by the KAS method. It is well known that activation energy can be used to predict the lifetime of materials. Using the E_a_ obtained for a conversion degree of 0.05, the material lifetime (t_T_) was calculated by analyzing the lifetime of rubber at various operating temperatures (T_T_). This conversion degree value might cause a substantial drop in the mechanical properties of a material. The proposed thermal lifetime calculation employed Equation (10):(10)$${\log(t}_{T})\;=\;\frac{E_{a}}{2{.303\mathrm{RT}}_{T}}\;+\;\log\left(\frac{E_{a}}{R\beta }\right)\;-\;\alpha$$
where t_T_ is the estimated thermal time to failure for a constant degree of conversion; T_T_ is the operating temperature; E_a_ is the activation energy for 0.05 conversion degree; and α is a tabulated value that is determined using E_a_/RT_c_, calculated from the numerical integration table given in reference [34]. T_c_ is the temperature for a 0.05 conversion degree at a heating rate of 20 °C/min.

## 3. Results and Discussion

LNR−g−MMA can be used as a compatibilizer in PLA/PP blends, as shown in Figure 3. LNR−g−MMA had functional groups with high polarity and reactivity. The terminal hydroxyl and carboxyl groups of the PLA matrix can react with the ester group of MMA on one side (Figure 3a) and attach between the isoprene portion of LNR−g−MMA and the PP matrix by the non−polar physical interaction on another side (Figure 3b). Additionally, Pangrahi et al. [35] prepared thermoplastic elastomers (TPEs) based on non-polar isotactic polypropylene (i-PP) and polar epichlorohydrin rubber (ECR) in the presence of the ethylene-acrylic ester-maleic anhydride terpolymer (EAE−MA−TP) used as the compatibilizer. They reported that the i-PP/ECR (40:60 wt%) composite containing a compatibilizer at a concentration of 5 wt% possesses exceptional mechanical properties. The unique properties of TPEs based on i-PP and ECR in the presence of the E−AE−MA−TP compatibilizer were attributed to chemical interactions between the saturated ester of the maleic anhydride portions and ethyl acrylate portions of the E−AE−MA−TP compatibilizer and the C–O groups of ECR and physical interactions between the ethylene portion of the E-AEMA-TP compatibilizer and the i-PP chains. Moreover, Bijarimi et al. [36] investigated the preparation of PLA by melt blending with LNR and LLDPE. The poor compatibility of LLDPE with PLA revealed the poor mechanical properties of the PLA/LLDPE binary blend. LNR was performed as a third−phase polymer to reduce the phase boundary and improve the compatibility of PLA and LLDPE. Consequently, LNR developed physical interfacial interactions between PLA and LLDPE.

### 3.1. Morphology

The morphology of the tensile-fractured surfaces of the PLA, PP, and PLA/PP (70:30 wt%) blends with various contents of LNR−g−MMA was studied using SEM, as shown in Figure 3. In Figure 3a, PLA showed a brittle fracture, while PP showed a ductile fracture and whitening deformation in the micrograph spread over the entire surface (Figure 3b). It is generally known that the incorporation of PP into the PLA matrix results in immiscibility due to the different polarity of the two polymers, and it was found that phase separation occurred. PP dispersed in the form of small balls is distributed evenly in the PLA matrix, showing an incompatible blend morphology. The small holes can be seen as a result of the sample being pulled apart by the tensile force, which also caused PP particles to separate from the PLA matrix, as shown in Figure 3c. Therefore, adding LNR−g−MMA to the mixture enhanced the miscibility and reduced the interfacial tensile between PLA and PP. The PMMA component in the grafting shell had a good miscibility with the PLA phase [23], and the LNR component compatibilized PLA/PP system [2], which improved compatibility. The result in the droplet size of the PP particles of the PLA/PP blends in the presence of LNR−g−MMA was smaller than that of the PLA/PP blend without LNR−g−MMA. The droplet size of the dispersed PP particles decreased significantly when the LNR−g−MMA was increased, as shown in Figure 3d–f. However, the morphology of the PLA/PP blends with the LNR−g−MMA of 10 phr presented tiny PP particles and was almost homogenous, as shown in Figure 3f.

### 3.2. FTIR Spectroscopy

Using FT−IR analysis, as shown in Figure 4, the functionalities of PP, PLA, and PP/PLA blends with and without a compatibilizer were examined. The transmittance bands of PLA (Figure 4a) at 1746, 1178, and 1084 cm^−1^ referred to C=O stretching, symmetric C–O–C stretching, and asymmetric CH_3_, respectively. The transmittance bands of PP (Figure 4b) at 2918–2835, 1457, and 1375 cm^−1^ were assigned to C–H stretching, –CH_3_ bending, and C–H bending, respectively. For PLA/PP blends of ratios 70:30 wt%, the transmittance bands that represent PP and PLA were observed in the polymer blends around 2918–2835, 1746, 1457, 1375, 1178, and 1084 cm^−1^ (Figure 4c,d). However, the characteristic spectrum of the ester linkage of (Figure 4e) LNR−g−MMA between 1729 and 837 cm^−1^ was not observed.

### 3.3. Mechanical Properties

#### 3.3.1. Tensile Properties

The tensile properties of neat PLA, neat PP, and PLA/PP (70:30 wt%) blends with different contents of LNR-g-MMA are presented in Figure 5a–c. In general, the stress–strain curves (Figure 5a) were highly dependent on the blend compositions. The tensile strength of the PLA and PP and the PLA/PP blend with LNR−g−MMA compatibilizer are shown in Figure 5b. It was found that pure PLA and PP have a tensile strength of 55.31 and 31.61 MPa, respectively. The tensile strength of pure PLA was much higher than that of pure PP due to the toughness of PP. Compared to pure polymers, the lower tensile strength in the blends may be caused by an incompatibility between non-polar PP and polar PLA because the applied stress could not be effectively transferred through the interface of the blended component. For the PLA/PP (70:30 wt%) blends with LNR−g−MMA, the tensile strength of the blends increased as the LNR−g−MMA content was increased. The addition of LNR−g−MMA to the PLA/PP blends improved the miscibility of the PMMA component as the grafting shell in the PLA phase and the yield strength due to the elastomeric nature of the LNR component as a core in the LNR−g−MMA particles.

Figure 5c illustrates that pure PP had a much better elongation at break than pure PLA due to the toughness of PP. Adding PP to the PLA matrix increased the elongation at break in the blend. This result implied that blends transitioned from brittle to ductile failure with the addition of PP. Compared to the blend without LNR−g−MMA, the binary blend system with LNR−g−MMA added demonstrated an increase in the elongation at break. However, the elongation at break increased gradually as the amount of LNR−g−MMA increased. This research found that adding LNR−g−MMA to the blend system increased the blend system’s tensile strength and elongation at break.

#### 3.3.2. Izod Impact Test

The impact strength of pure PLA and PP and PLA/PP (70:30 wt%) blend without and with LNR−g−MMA compatibilizer is shown in Figure 5d. The impact strength of the pure PLA and PP show an impact strength of 2.75 and 3.64 kJ/m^2^, respectively, indicating that pure PLA is much more brittle compared to virgin PP. In the PLA/PP (70:30 wt%) blend without LNR−g−MMA compatibilizer, the value decreased slightly to 3.09 kJ/m^2^. PLA and PP have a polar difference, in which polymer blends occur a phase separation. However, impact strength of the polymer blends was slightly increased from 3.14 to 3.57 kJ/m^2^ with an increase of the LNR−g−MMA content from 2.5 to 10.0 phr, as shown in Figure 5d. This can be concluded by the presence of LNR−g−MMA, which acted as an impact modifier in the PLA/PP blends due to the elastomeric nature of the LNR component as a core in the LNR−g−MMA particles, as discussed earlier. A similar observation was found by Bijarmi et al. [2]. They reported that PLA toughness could be improved by direct melt blending with PP and LNR compatibilizer, which contributed to the enhanced elongation at break, impact strength, and flexural properties.

#### 3.3.3. Water Absorption

The water absorption of pure PLA and PP and PLA/PP (70:30 wt%) blend without and with LNR−g−MMA compatibilizer is shown in Figure 5e. It was found that pure PLA has a much higher absorption percentage than pure PP. Pure PLA is a polar molecule and a hydrophilic material consisting of hydroxyl groups that form intermolecular hydrogen bonds with the water molecules, resulting in absorbed high water in the pure PLA. Figure 5e shows that the PLA/PP (70:30 wt%) blend without LNR−g−MMA compatibilizer absorbed water more than that of pure PLA. In general, the incorporation of PP into the PLA matrix exhibits interfacial regions which occur in many micropores in the polymer blend due to an immiscible system with a two−phase morphology. The micropores between the two phases allow for the absorption of water. When the compatibilizer was added, the blend became a more uniform dispersion of PP throughout the PLA matrix, reducing the number and size of the micropores, and preventing water from reaching the polymer chains. Consequently, the water absorption percentage decreased as the LNR−g−MMA content increased when it was added to a PLA/PP blend. The LNR−g−MMA made the PLA/PP composite more miscible, reduced the number and size of micropores, and homogenized and refined the matrix [37,38].

### 3.4. Thermal Analysis

#### 3.4.1. Thermal Stability

According to ASTM E 1877 [39], which provides the necessary mathematical processes employed in determining activation energy and lifetime, TGA analyses were carried out, followed by the evaluation of the thermal stability, kinetics of degradation, and lifetime of the PLA, PP, and PLA/PP blends. All polymers were heated at rates of 10, 20, 30, and 40 °C/min, between 50 and 800 °C in an atmosphere of air. The TG and DTG curves for the neat polymers and polymer blends of PLA/PP ratios 70:30 wt%, and LNR−g−MMA with a content of 0.0, 2.5, 5.0, and 10.0 phr as a compatibilizer used in the current study, are shown in Figure 6 and Figure 7 for the four used heating rates. Both neat polymers showed single−stage decomposition as shown in Figure 6a,b. Unlike the neat PP and PLA, the two−step degradation process in immiscible polymer blends was clearly seen in PP/PLA blends, as shown in Figure 6c–f. The DTG curves (Figure 7c–f) identified two decomposition peaks. More notably, the first peak was related to the decomposition of the blend’s PLA, while the second peak, which appeared at a higher temperature, is associated with the degradation of PP [40]. The decomposition temperature shifted toward a higher temperature with the increase in heating rate. The TG data showing the thermal degradation temperatures of the neat polymers and polymer blends of PLA/PP ratios 70:30 wt%, using an LNR−g−MMA content of 0.0, 2.5, 5.0, and 10.0 phr as a compatibilizer, are listed in Table 1. The initial temperature (T_i_) and the final temperature (T_f_) were obtained with a bitangent method, and T_m_ is the temperature at the maximum weight loss rate, which can be obtained from the peak of the DTG at different heating rates (Figure 7). All blends under investigation showed an exothermic effect consistent with the thermal degradation process. Compared to the neat PP, the thermal decomposition curve of neat PLA shifted towards low temperatures, indicating that PLA was less thermally stable than PP. The incorporation of PP significantly changed the degradation temperature trends. Blending PLA with PP (70:30 wt%) increased the blend’s thermal stability or made it more resistant to temperature compared to the pure PLA, but the addition of a compatibilizer decreased the blend’s thermal stability by increasing interfacial adhesion and interaction between polymer phases.

#### 3.4.2. Activation Energy

The activation energies were determined by the KAS method using the TGA curves (Figure 6) at various heating rates. In Figure 8, ln(β/T^2^) as a function of 1000/T is plotted at different conversion degrees, varying from 0.025 to 0.5. As can be observed from Figure 8, the results showed parallel linear straight lines fitted to the experiment that validated the model. From the slope of the resulting straight line, the activation energy (E_a_) of the degradation process was calculated. According to these results, the activation energy increases up to 0.2 conversion degrees, followed by a slight increase. This behavior can be better visualized in Figure 9, plotting the E_a_ versus the conversion degrees. The calculated values that were restricted to the degrees of conversion range of 0.025–0.2 with intervals of 0.025 were trustworthy, but the results outside of this range were merely intended to serve as a reference [30]. The E_a_ indicates how rapidly the reaction takes place. The E_a_ (w = 0.05) values of PLA and PP were 108.5 and 156.4 kJ/mol, respectively. Therefore, the higher E_a_ value of PP degradation compared to that of PLA presented a higher thermal stability. The addition of PP to the PLA matrix increased the value of E_a_ (w = 0.05) up to 144.9 kJ/mol. In comparison, the E_a_ values (w = 0.05) of PLA/PP/LNR−g−MMA (70:30:2.5 phr), PLA/PP/LNR−g−MMA (70:30:5.0 phr), and PLA/PP/LNR−g−MMA (70:30:10.0 phr) were 134.3, 125.5, and 109.8 kJ/mol, respectively. The addition of PP in PLA increased the activation energy due to the incompatibility between the two polymers. Comparatively, adding a compatibilizer improved the compatibility of the two polymers and lowered the activation energy, decreasing the blends’ thermal stability.

#### 3.4.3. Lifetime Estimation

Figure 10 shows the thermal lifetime behavior estimated for PLA, PP, and PLA/PP (70:30 wt%) blends without and with LNR−g−MMA obtained for different failure temperatures. In this case, the expected lifetime was calculated based on the sample’s 0.05 conversion degree at a heating rate of 20 °C/min during thermogravimetry tests. The lifetime values of PLA, PP, and PLA/PP (70:30 wt%) blend without and with LNR−g−MMA for a temperature range between 25 and 150 °C were calculated using Equation (10) and are plotted in Figure 10. Regardless of the method employed for estimation, it is obvious that the lifetime of all blends was temperature dependent and was exponential with an increase in temperature. The lifetime estimation, determined using a conversion degree failure of 0.05 at a heating rate of 20 °C/min, of PLA, PP, and the PLA/PP (70:30 wt%) blend with LNR−g−MMA of 0.0, 2.5, 5.0, and 10.0 phr was 2.95 × 10^3^, 3.37 × 10^10^, 3.54 × 10^7^, 2.30 × 10^6^, 2.99 × 10^5^, and 6.23 × 10^3^ years at 25 °C and 1.00 × 10^2^, 2.56 × 10^8^, 3.85 × 10^5^, 3.48 × 10^4^, 5.95 × 10^3^, and 2.03 × 10^2^ years at 50 °C, respectively. The increase in temperature tended to diminish the durability of the material. Consequently, the temperature affected the entire lifetime of the material. Because a lifetime is based on activation energy, PP had a longer lifetime than PLA and PLA/PP (70:30 wt%) blend. The PLA/PP (70:30 wt%) blend without LNR−g−MMA had a longer lifetime than PLA/PP (70:30 wt%) blends with LNR−g−MMA content. This study showed how activation energy affected the thermal performance of PLA/PP/LNR−g−MMA blends, significantly shortening their lifetime.

## 4. Conclusions

This study aimed to investigate the effect of LNR−g−MMA content on the morphology, mechanical properties, water absorption, thermal degradation, and a lifetime of blends based on PLA and PP. PLA polymer, PP polymer, and PLA/PP (70:30 wt%) blend, with LNR−g−MMA as compatibilizers with a content of 0.0, 2.5, 5.0, and 10.0 phr, respectively, were prepared by internal mixing and compression molding. SEM revealed that the PLA/PP blend underwent phase separation, resulting in the dispersion of PP in the form of small balls distributed evenly in the PLA matrix. When the presence of LNR−g−MMA increased, the PP droplet sizes reduced considerably, with an almost homogenized and refined blend morphology. LNR−g−MMA was added as a compatibilizer to improve the compatibility between PLA and PP. The neat PLA and PP has a tensile strength of 55.31 and 31.61 MPa and an elongation at break of 3.85 and 19.60%, respectively. However, an increasing trend of tensile strength and elongation at break of the polymer blends was observed, slightly increasing from 25.78 to 29.35 MPa and from 3.59 to 5.56%, respectively, with an increase in the LNR−g−MMA content from 0.0 to 10.0 phr, respectively. PLA displayed an impact strength of 2.75 kJ/m^2^, indicating a brittle polymer. In the PLA/PP (70:30 wt%) system, an increasing trend was observed for the impact strength of the polymer blends, which slightly increased from 3.09 to 3.57 kJ/m^2^ as the LNR−g−MMA content increased from 0.0 to 10.0 phr. Overall, the results showed that the values of tensile strength, elongation at break, and impact strength of the polymer blends increased, whereas water absorption values decreased with an increase in LNR−g−MMA content. Thermal degradation kinetics was studied over a temperature range of 50–800 °C, at multiple heating rates of 10, 20, 30, and 40 °C/min. When PLA was blended with PP at 70:30 wt%, the blend was more stable or resistant to temperature than PLA. However, the addition of LNR−g−MMA as a compatibilizer decreased the blend’s thermal stability by increasing interfacial adhesion and interaction between the polymer phases. The above results suggested that adding PP to the PLA matrix increased the activation energy. Comparatively, adding a compatibilizer enhanced the compatibility of two polymers and decreased activation energy, thus decreasing the blends’ thermal stability. Due to the lifetime calculated from the E_a_ values of the polymers, PP was more durable than PLA and the PLA/PP (70:30 wt%) blend. The PLA/PP (70:30 wt%) blend without LNR−g−MMA had a longer lifetime than those with LNR−g−MMA. A material’s thermal stability can influence the processing and applications of polymeric materials, thus necessitating knowledge of their thermal lifetime under a particular end-point criterion and operating temperature.

## Data Availability

All the data is included in the article.

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
