# Peer review of "Effect of LNR-g-MMA on the Mechanical Properties and Lifetime Estimation of PLA/PP Blends"

_polymers, 2023, doi:10.3390/polym15071712_

Round 1

Reviewer 1 Report

The paper "Effect of LNR-g-MMA on the Mechanical Properties and Lifetime Estimation of PLA/PP Blends" shows interesting results where a new compatibilizer was used for the fabrication of the PLA/PP blends. The paper can be accepted for publication after major revision. The following issues should be clarified.

Why was LNR-g-MMA chosen as a compatibilizer for the fabrication of the PLA/PP blends? It should be explained exactly in the introduction.

What is DRC (line 139)? What exactly is the Emulvin WA, and what does it do? Please add a scheme that illustrates the preparation of LNR-g-MMA.

Fig. 5b is not described in the text. The caption for figure 5 should be rephrased.

Please include a scheme demonstrating the role of LNR-g-MMA as a compatibilizer in PLA/PP blends.

It is not clear to me why water absorption is essentially less for samples with the increased context of the LNR-g-MMA.

I suggest citing the paper where similar systems were developed to improve the compatibility of polypropylene with hydrophilic polymers or fillers:

https://doi.org/10.1002/masy.200450638

https://doi.org/10.3390/polym14235253

Author Response

Author's Response to Decision Letter for

Effect of LNR-g-MMA on the Mechanical Properties and Lifetime Estimation of PLA/PP Blends (polymers-2254747)

Dear Reviewer,

We thank the Reviewer for taking the time and effort to improve the manuscript's quality. Our responses to the Reviewer’s comments are described below:

  Response to Reviewer 1 Comments

The paper “Effect of LNR-g-MMA on the Mechanical Properties and Lifetime Estimation of PLA/PP Blends” shows interesting results where a new compatibilizer was used for the fabrication of the PLA/PP blends. The paper can be accepted for publication after major revision. The following issues should be clarified.

1.

Why was LNR-g-MMA chosen as a compatibilizer for the fabrication of the PLA/PP blends? It should be explained exactly in the introduction.  

Response: “The preparation of liquid natural rubber (LNR) from the degradation of natural rubber (NR) latex is related to the breaking of the natural rubber (NR) chain via direct scission of the C-C or C=C bonds of the polyisoprene backbone. Moreover, LNR with a similar microstructure of NR having a molecular weight below 20,000 g/mol becomes an attractive material for enhancement of further modifications. The graft copolymers consist of one or more side chains structurally distinct from the main polymer chains or backbone. and Graft copolymer synthesis has been performed in solution [24], solid [25], and latex phases [26]; however, latex modification may be the most cost-effective and practical method. The graft copolymer of methyl methacrylate onto LNR (LNR-g-MMA), comprising an inner soft polymer sphere (LNR), the “core” and an outer rigid polymer (PMMA), the “shell” can be expected to have better impact-resistance properties” was added.

(Line 101-106 and 112-117)

2.

What is DRC (line 139)? What exactly is the Emulvin WA, and what does it do? Please add a scheme that illustrates the preparation of LNR-g-MMA.

Response:

2.1 (dry rubber content)

2.2 (C24H22O2) used as a surfactant

2.3 Scheme 1 showed oxidative degradation in producing hydroxyl-terminated LNR. Free radicals formed from the bond cleavage will lead to the degradation of NR.

Scheme 1. Proposed mechanism of preparing LNR.

The proposed mechanism of preparing LNR-g-MMA initiated with cumene hydroperoxide and tetraethylene pentamine was shown in Scheme 2. The cumyloxyl radicals (generated from cumene hydroperoxide) tend to favor the abstraction of α-methylenic hydrogen from the allylic carbon in the polyisoprene backbone that leads to the formation of polyisoprene radicals initiating MMA to form the graft copolymers.

Scheme 2. Proposed mechanism of grafting of LNR with MMA via radical polymerization.

(Line 171, 172, 177-181, and 191-197)

3.

Fig. 5b is not described in the text. The caption for figure 5 should be rephrased.

Response:

5.1 “Figure 5(c) illustrates that pure PP had a much more excellent elongation at break than pure PLA due to the toughness of PP. Adding PP to the PLA matrix increased the elongation at break in the blend. This result implied that blends transitioned from brittle to ductile failure with the addition of PP. Compared to the blend without LNR-g-MMA, the binary blend system with LNR-g-MMA added demonstrated an increase in elongation at break. However, the elongation at break increased gradually as the amount of LNR-g-MMA increased. This research found that adding LNR-g-MMA to the blend system increased the blend system's tensile strength and elongation at break.” Was added

(Line 399-406)

5.2 “Figure 5. Physical properties of PLA, PP, and PLA/PP (70;30) blends with LNR-g-MMA of 0.0, 2.5, 5.0, and 10.0 phr: (a) stress-strain curves, (b) tensile strength, (c) elongation at break, (d) impact strength, and (e) water absorption. The error bars indicate the standard deviation.” was changed.

(Line 409-411)

4.

Please include a scheme demonstrating the role of LNR-g-MMA as a compatibilizer in PLA/PP blends.

Response:

“LNR-g-MMA can be used as a compatibilizer in PLA/PP blends, as shown in Scheme 3. LNR-g-MMA had functional groups with high polarity and reactivity. The terminal hydroxyl and carboxyl groups of the PLA matrix could react with the ester group of MMA on one side and attract with the PP matrix by hydrogen bond on another side.

Scheme 3. Possible reaction mechanism of PLA/PP blends in the presence of LNR-g-MMA.” was added.

(Line 333-340)

5.

It is not clear to me why water absorption is essentially less for samples with the increased context of the LNR-g-MMA.

Response: “In general, the incorporation of PP in the PLA matrix has interfacial regions which occur in many micropores in the polymer blend due to an immiscible system with a two-phase morphology. The micropores between the two phases allow for the absorption of water. When the compatibilizer was added, the blend became more uniform dispersion of PP throughout the PLA matrix, reducing the number and size of the micropores, and preventing water from reaching the polymer chains. Consequently, the water absorption percentage decreased as LNR-g-MMA content increased when it was added to a PLA/PP blend. The LNR-g-MMA made the PLA/PP composite more miscible, reduced the number and size of micropores, and homogenized and refined the matrix [32,33].” was added.

(Line 436-445)

6.

I suggest citing the paper where similar systems were developed to improve the compatibility of polypropylene with hydrophilic polymers or fillers:

https://doi.org/10.1002/masy.200450638

https://doi.org/10.3390/polym14235253

Response: “https://doi.org/10.3390/polym14235253” was added.

(Line 113)

Reviewer 2 Report

Dear Author,

I studied your manuscript entitled "Effect of LNR-g-MMA on the Mechanical Properties and Lifetime Estimation of PLA/PP Blends". In my opinion, your paper was written hastily, and some spaces need to be improved in terms of journal quality. I recommend a deep revision before further consideration for publication in the Polymers.

1) The quality of the abstract and conclusion should be enhanced by the inclusion of significant research findings. More quantitative information in these sections would be beneficial.

2) The introduction fails to point out the novelty of the work and its contribution to the state-of-the-art. The authors should clearly state the objectives of their experiments and the motivations driving their research in the first place. Furthermore, this manuscript has a phenomenological style, observing a result and explaining it with statements. It would be helpful if you conducted more analysis based on published research.

3) How did you select the PLA/PP ratio and LNR-g-MMA content? The authors should spend a few words on this matter. 

4) There is a lack of references to recent works. Only 3 of the 27 references are within the last three years. The authors have to complete their investigation taking a look at the up-to-date references to better clarify the innovative finding described. Some related papers were also strongly suggested to be cited in the references:

a) https://doi.org/10.3390/polym15040952

b) https://doi.org/10.1177/08927057221118823

c) https://doi.org/10.3390/polym14194205

5) Some more analyses should be reported and discussed. For example, the results of WAXD analysis could be reported and discussed to determine the crystallographic properties of the samples.

6) PLA is subjected to facile thermal and mechanical degradation during sample preparation. Can the authors assume (with experimental support) that their results are free of these problems?

7) All stress-strain curves should be presented (associated with Figure 5).

8) English language needs some polishing since some terms are vague.

Author Response

Author's Response to Decision Letter for

Effect of LNR-g-MMA on the Mechanical Properties and Lifetime Estimation of PLA/PP Blends (polymers-2254747)

Dear Reviewer,

We thank the Reviewer for taking the time and effort to improve the manuscript's quality. Our responses to the Reviewer’s comments are described below:

Response to Reviewer 2 Comments

Dear Authors,

I studied your manuscript entitled “Effect of LNR-g-MMA on the Mechanical Properties and Lifetime Estimation of PLA/PP Blends”. In my opinion, your paper was written hastily, and some spaces need to be improved in terms of journal quality. I recommend a deep revision before further consideration for publication in the Polymers.   

1.

The quality of the abstract and conclusion should be enhanced by the inclusion of significant research findings. More quantitative information in these sections would be beneficial.

Response: “The neat PLA and PP, the tensile strength of 55.31 and 31.61 MPa and elongation at break of 3.85 and 19.60%, respectively. However, an increasing trend of tensile strength and elongation at break of polymer blends was slightly increased from 25.78 to 29.35 MPa and from 3.59 to 5.56% with an increase of the LNR-g-MMA content from 0.0 to 10.0 phr, respectively. PLA displayed an impact strength of 2.75 kJ/m2, indicating a brittle polymer. In the PLA/PP (70:30) system, an increasing trend of impact strength of polymer blends was slightly increased from 3.09 to 3.57 kJ/m2 with an increase of the LNR-g-MMA content from 0.0 to 10.0 phr.” was added.

(Line 556-563)

2.

The introduction fails to point out the novelty of the work and its contribution to the state-of-the-art. The authors should clearly state the objectives of their experiments and the motivations driving their research in the first place. Furthermore, this manuscript has a phenomenological style, observing a result and explaining it with statements. It would be helpful if you conducted more analysis based on published research. 

Response: “Overall due to the brittleness of PLA, it can be blended with PP at the ratios of 70:30 wt% [20,21], yielding higher mechanical properties. It is well known that even a small quantity of compatibilizer can cause significant changes to a material's mechanical properties [19]. LNR-base latex particles with a sufficiently thick shell, comprising an inner soft polymer sphere, the “core” and an outer hard polymer (PMMA), the “shell,” give a free-flowing powder, which is usually employed in continuous extrusion processes for the preparation of polymer blends.

This work aims to improve the compatibility between two types of polymer blends, PLA/PP (70:30 wt%), by adding a third component of LNR-g-MMA to act as the compatibilizer agent. The first portion of this study focused on synthesizing LNR-g-MMA using a redox initiator for core-shell shaped particles. Later, the PLA/PP (70:30 wt%) blends with the amount of LNR-g-MMA content (0, 2.5, 5.0, and 10.0 phr) were prepared by melt blending in an internal mixer and molded by compression molding. The effect of LNR-g-MMA contents on the morphological, mechanical, water absorption, thermal degradation behavior, and thermal stability of PLA/PP blends was investigated. The model-free kinetic method was used to estimate the degradation activation energy (Ea), which was used to predict the lifetime of PLA, PP, and PP/PLA blends without and with LNR-g-MMA contents as a compatibilizer.” was changed.

(Line 140-157)

3.

How did you select the PLA/PP ratio and LNR-g-MMA content? The authors should spend a few words on this matter.

Response: “The relationship between immiscible PLA/PP blends of different compositions and the compatibilizer loading via melt mixing has been studied in a few works. The effect of compatibilizer loading on the blends' properties was also investigated. However, addition of compatibilizer more than 20-30% into PLA matrix can cause the phase between polymer blend to separate [19]. Kang et al. [20] prepared PLA/PP blends of different compositions with 0, 2.5, 5.0, 7.5, 10.0 wt% EBA-GMA (ethylene-butyl acrylate-glycidyl methacrylate terpolymer) used as a compatibilizer. They reported that the tensile strength of the PLA/PP (70:30) blends reached a maximum for 2.5 wt% EBA-GMA, and impact strength increased with increasing EBA-GMA content. Jariyakulsith and Puajindanetr [21] studied the relationship between compatibilizer and yield strength of PLA/PP blends at 70:30, 50:50, and 30:70 ratios. In addition, PP-g-MAH was added as a compatibilizer at 0.3 and 0.7 phr (parts per hundred) of PLA/PP resin and dicumyl peroxide (DCP) as an initiator at 0.03 and 0.07 phr. They discovered that a 70:30 blend of 0.3 phr PP-g-MAH and 0.03 phr DCP provided the highest yield strength of 27.68 MPa.” was added.

(Line 87-100)

4.

There is a lack of references to recent works. Only 3 of the 27 references are within the last three years. The authors have to complete their investigation taking a look at the up-to-date references to better clarify the innovative finding described. Some related papers were also strongly suggested to be cited in the references:

a) https://doi.org/10.3390/polym15040952

b) https://doi.org/10.1177/08927057221118823

c) https://doi.org/10.3390/polym14194205

Response: “a) https://doi.org/10.3390/polym15040952

and c) https://doi.org/10.3390/polym14194205 ” was added.

(Line 47)

5.

Some more analyses should be reported and discussed. For example, the results of WAXD analysis could be reported and discussed to determine the crystallographic properties of the samples.

Response: We did not execute the WAXD analysis.

6.

PLA is subjected to facile thermal and mechanical degradation during sample preparation. Can the authors assume (with experimental support) that their results are free of these problems?

Response: Yes, we can assume (with experimental support) that these results are free of these problems because of constant blend ratio.

7.

All stress-strain curves should be presented (associated with Figure 5).

Response: “the stress-strain curves” were added.

(Line 407 (Figure 5(a))

8.

English language needs some polishing since some terms are vague.

Response: English language was checked and corrected.

Round 2

Reviewer 1 Report

The paper was essentially improved after revision but some minor issues should be clarified.

First, the mechanism of the decomposition of the cumene hydroperoxide should be clarified (https://doi.org/10.1002/aic.11487) It is well-known for the decomposition of cumene hydroperoxide, the first radical is cumyloxyl radical, and the second - hydroxyl radical (*OH). Scheme 2 should be corrected. Appropriate references to scheme 1 should be added.

Scheme 3 and the explanation of this scheme in the text are hard for understanding.  A suggested hydrogen bond is impossible to form since the scheme and text should seriously be proceeded before the paper will be accepted for publication. Appropriate references to scheme 3 should be added.

Author Response

Author's Response to Decision Letter for

Effect of LNR-g-MMA on the Mechanical Properties and Lifetime Estimation of PLA/PP Blends (polymers-2254747)

Dear Reviewer,

We thank the Reviewer for taking the time and effort to improve the manuscript’s quality. Our responses to the Reviewer’s comments are included in an attached file. Please find the enclosed.

Yours faithfully,

Dear Editor,

We thank the Editor for taking the time and effort to improve the manuscript’s quality. Our responses to the Reviewer’s comments are included in an attached file. Please find the enclosed.

Yours faithfully,
